# The Effect of Curcumin on Renal Ischemia/Reperfusion Injury in Diabetic Rats

**DOI:** 10.3390/nu14142798

**Published:** 2022-07-07

**Authors:** Douglas Ikedo Machado, Eloiza de Oliveira Silva, Sara Ventura, Maria de Fatima Fernandes Vattimo

**Affiliations:** School of Nursing, University of São Paulo, São Paulo 05403-000, Brazil; ikedo13@alumni.usp.br (D.I.M.); sara.ventura@usp.br (S.V.); nephron@usp.br (M.d.F.F.V.)

**Keywords:** diabetes, curcumin, ischemia/reperfusion

## Abstract

Chronic kidney disease (CKD) and acute kidney injury (AKI) are global health problems that affect over 850 million people, twice the number of diabetic individuals around the world. Diabetes mellitus (DM) is known to increase the susceptibility to AKI. Plants and foods, such as curcumin, are traditionally used as treatments for various diseases due to its wide range of bioactive compounds that exert antioxidative, anti-inflammatory, antimicrobial and anticancer properties. The aim of this study is to evaluate the effect of curcumin in diabetic rats with AKI. Adult male Wistar rats, weighing between 250 and 290 g, were randomized into four groups: Citrate (citrate buffer, i.v., single dose, on Day 1 of the protocol); DM (streptozotocin (STZ), 65 mg/k, single dose, i.v., on Day 1); DM + I/R (DM rats that, on Day 26, had the renal pedicle clamped for 30 min on both sides); DM + I/R + Curcumin (DM + I/R rats submitted to curcumin treatment). Results showed that IR worsened renal function and oxidative stress in DM rats, but the DM + IR + Curcumin group showed an increase in inulin clearance and a decrease in serum creatinine and in NGAL, in addition to an improvement in renal hemodynamics. These effects were accompanied by a reduction in oxidative and nitrosative metabolites and an increase in the thiol antioxidant reserve when curcumin was administered to the DM + IR group.

## 1. Introduction

It is known that diabetes is an important risk factor for acute kidney injury (AKI) [1], whose incidence is higher in surgeries in diabetic patients [2]. AKI induced by renal ischemia/reperfusion and DM are prevalent complications in clinical routine and share common pathophysiological mechanisms. One of the main complications of DM in individuals with sustained hyperglycemia is diabetic kidney disease (DKD), also known as diabetic nephropathy (DN), present in approximately 30% to 50% of patients [3]. Aside from renal replacement and supportive treatment, there are few specific therapeutics available to treat DKD [4]. Therefore, a new therapeutic strategy is urgently needed to preserve renal impairment and functional decline after AKI.

Clinically, DN is characterized by reduced glomerular filtration rate (GFR) and proteinuria (urinary albumin excretion), which is most often associated with an increase in blood pressure [3]. The long-term complications from DM are classified as microvascular (nephropathy, retinopathy and neuropathy) or macrovascular (cerebrovascular accident and atherosclerosis) [5].

The pathogenic effects of hyperglycemia result from the overproduction of superoxide by the mitochondrial electron transport chain, which results in oxidative stress [5]. Sustained hyperglycemia induces the metabolism of excess glucose, promoting the formation of advanced glycation end products (AGEs) and production of reactive oxygen species (ROS), mainly through the hexosamine and polyol pathways, changing mediators of vasoconstriction (prostaglandin) and vasodilation (nitric oxide), that result in hemodynamic insult with reduced renal blood flow [6]. Thus, the production of proinflammatory cytokines initiates the inflammatory process associated with the redox processes, perpetuating the production of ROS [6].

ROS can cause numerous detrimental effects that induce and aggravate diabetes, including the reduction in glucose transport channels and insulin secretion, protein fragmentation and oxidation, DNA damage, free fatty acid generation and increase in vascular permeability. Furthermore, ROS causes an aggravation in renal function, causing podocyte damage, decreased GFR, proteinuria, tubulointerstitial fibrosis and changes in renal hemodynamics [2].

The simultaneity of these events results in glomerulosclerosis and tubulointerstitial injury, increasing susceptibility to AKI caused by ischemia and reperfusion (I/R) in diabetic individuals [6,7].

Turmeric Long Linn, popularly known as turmeric, saffron da terra, yellow ginger, un root or curcumin, is commonly utilized as a culinary spice. This type of plant is characterized by orange tuberous rhizomes and is widely cultivated in Southeast Asia, where it has been used as a natural therapeutic remedy for various pathological conditions since ancient times [8]. Curcumin has shown to be effective in the regulation of glycemia and lipidemia [9] with an action mechanism similar to antidiabetic medications [10].

In addition, recent studies have demonstrated promising effects of curcumin for the treatment of autoimmune diseases, such as type 1 diabetes [8]. It has been also demonstrated that it could have a potential role in preventing and treating several diseases due to its antibacterial, antiviral, anti-ischemic, hepatoprotective, nephroprotective, antirheumatic and anticancer activities [8,11,12].

The renal benefits of phytomedicines such as Justicia acuminatissima, Uncaria tomentosa and isoflavone have been shown in I/R experimental models [13,14,15]. Thus, the hypothesis of this study is that curcumin can exert a renoprotective effect in the situation of acute insult due to ischemia, suppressing the morbidity of DM when in coexistence. The aim of this study is to evaluate the effect of curcumin on renal function, hemodynamics and oxidative profile of diabetic rats submitted to I/R AKI.

## 2. Materials and Methods

Adult male Wistar rats, weighing between 250 and 290 g were used for this study. All rats had free access to food and water and were housed in a temperature-controlled facility with 12 h light/dark circles [16].

DM was induced by administering 65 mg/kg intravenous (i.v.) through flow of streptozotocin (STZ) (Sigma Chemical Company, St Louis, MO, USA) diluted in 0.1 M citrate buffer at a pH of 4.2. Animals with blood glucose levels higher than 250 mg/dL 48 h after the induction procedure were considered diabetic [6].

Curcumin was administered for 10 days, starting on the 17th day of the experimental protocol, with a dosage of 30 mg/kg/day, orally, diluted in a 0.5% carboxymethylcellulose solution [12].

AKI I/R rats were anesthetized with Thiopentax^®^ (sodium thiopental: 50 mg/kg, i.p.) and submitted to laparotomy for bilateral clamping of the renal pedicles for 30 min with nontraumatic vascular clamps. All animals were evaluated during anesthetic recovery and received analgesic post procedure (tramadol: 15 mg/kg, i.p., 8/8 h) [14].

Animals (*n* = 20) used in this study were randomly allocated into four groups: (1) Citrate group: animals that received citrate buffer (STZ diluent) at a pH of 4.2, i.v., single dose, on the 1st day of the experimental protocol; (2) DM group: STZ (65 mg/kg) on the 1st day of the experimental protocol; (3) DM + I/R group: DM rats that, on the 26th day, had bilateral clamping of the renal pedicles performed; (4) DM + I/R + Curcumin: DM rats that, on the 17th day, started receiving curcumin 30 mg/kg/day until the 27th day, and, on the 26th day of the protocol, bilateral clamping of the renal pedicles was performed.

The experimental protocols lasted 4 weeks (28 days), during which the animals’ body weight and blood glucose were measured weekly. On the 27th day of the experimental protocol, the animals from the different groups were placed in metabolic cages to measure the urinary volume, water intake and 24 h food intake. They were removed from the metabolic cages on the 28th day and were then anesthetized and submitted to the necessary procedures to analyze the renal function using the technique of determination of inulin clearance. In order to perform the procedure, the animals were submitted to a laparotomy and terminal blood collection through puncture of the abdominal aorta. The right kidney was removed and then weighed to calculate the kidney weight/animal weight ratio, and the left kidney was removed, packaged and stored in a refrigerator at −80 °C for further studies to measure nonprotein thiols. At the end of the experiment, the animals were euthanized by terminal blood collection, according to the ethical standards for handling animals in a research laboratory [15].

The food consumption and body weight were measured weekly using an analytical scale, and the results were expressed in grams. Quantitative determination of blood glucose was performed on a fresh capillary blood sample collected from a tail puncture, and values were obtained using test strips and a blood glucose monitor (Accu Check^®^ Active, Roche Diagnostics, São Paulo, Brazil.). For the four subsequent weeks, body weight and blood glucose were monitored weekly.

Glomerular filtration rate was determined using the inulin clearance technique. The catheterization of the jugular vein was performed for inulin infusion, and an initial dose of 100 mg/kg corporal weight of diluted inulin was administered, followed by a continuous infusion of 10 mg/kg corporal weight for 2 h of the experiment, at a rate of 0.04 mL/min. After a stabilization period of 30 min, urine collection was performed every 30 min through bladder catheterization, and blood samples were collected every 60 min for analysis of urinary and plasma inulin concentration using the Antrona method. Inulin clearance was expressed in mL/min/100 g [17,18].

The measurement of urinary and serum creatinine was determined by the colorimetry method known as the Jaffé method [19].

The dosage of NGAL was carried out in the Experimental Laboratory of Animal Models (LEMA) using ELISA kit standards (NGAL ELISA kit, BioVendor, research and diagnostic products, Ref: RD 191102200R) [20].

The left renal artery was isolated and involved in an ultrasonic probe to measure renal blood flow (RBF). Mean arterial pressure (MAP) was recorded through carotid artery catheterization to assess renal vascular resistance (RVR), calculated using the formula RVR = MAP/RBF [18].

Peroxides are considered to be potential indicators of the formation of reactive oxygen molecules and were used to evaluate the oxidative profile. They are found in all bodily fluids, especially urine. Changes in levels are considered markers for H2O2 generation or predictors of the extent of oxidative damage in vivo. Direct measurement of peroxides can be performed using the FOX-2 analysis method [21]. The FOX-2 method consists of determining peroxide levels using the iron–xylenol orange method. The orange xylenol [acid (o-cresolsulfonaphthaline 3′, 3″ -bis (methylamino) diacetic acid] has high selectivity for the Fe3 + ion, producing a bluish-purple color complex (α= 4.3 × 10^4^ M^−1^ cm^−1^) (Vetec Química-RJ, Brazil). The reading was performed by spectrophotometry at an absorbance of 560 nm after removing protein residues and other materials from centrifugation [21]. Values were stabilized per gram of urinary creatinine and expressed in nmol of peroxides/gram of creatinine.

The synthesis of nitric oxide (NO) was evaluated through the quantification of nitrate (NO3−, a stable metabolite of NO, using the Griess method. The result is colorimetric and based on the reaction of nitrites with sulphanilic acid and copulation with alpha-naphthylamine hydrochloride in an acidic medium (pH between 2.5 and 5.0), forming alpha-naphthylamine- *p*-azobenzene-*p*-sulfonic acid in the color pink. The reading was performed at absorbance of 545 nm in an ELISA reader. The absorbance of the samples was compared to a standard curve of sodium nitrate (NaNO^3^) at a concentration of 0.1 to 1.0 M [22]. The equation for measuring NO was adjusted for urinary creatinine values and all values obtained were stabilized in nmol of NO per gram of urinary creatinine.

MDA is one of the frequently analyzed aldehydes in quantitative and qualitative analytical methods to determine lipid peroxidation indexes and can be detected by several methods, including through the reaction with thiobarbituric acid, [23]. The amount of MDA (TBARs) detected in the samples in nmol was calculated using the 1.56 × 10^5^ M^−1^ cm^−1^ molar extinction coefficient. Values were expressed in nmol per gram of creatinine [23].

The most relevant thiol compound present in biological systems is glutathione (GSH). It is present in all cells and constitutes the main redox buffer, whose biological functions are centered on the thiol group, present in the side chain, which undergoes repeated oxidation and reduction cycles. High concentrations of GSSG indicate redox imbalance or the presence of oxidative damage. Therefore, the amount of thiols was used as an indicator of oxidative stress, considering the following principle: the greater the degree of oxidative stress, the higher the levels of oxidized thiols and the lower the concentrations of thiols in the renal tissue [24]. The correlation with the measurement of soluble thiols was made from the quantification of total proteins. All values obtained were stabilized in nmol of thiols/mg of total proteins.

Results are presented as mean ± standard deviation. Data were submitted to ANOVA variance analysis, followed by Tukey’s multiple comparisons test. Values of *p* < 0.05 were considered significant.

## 3. Results

The Citrate group presented results that were considered normal standards for the parameters analyzed below.

### 3.1. Physiological Parameters

As shown in Table 1, the diabetic groups DM, DM + I/R and DM + I/R + Curcumin had different blood glucose parameters when compared to the Citrate group during the 2nd week, 3rd week, 4th week. These groups presented a similar evolution in the mentioned parameter and maintained a linear difference with the Citrate group.

The DM group was the only one that showed a decrease in body weight during the four weeks and compared to the Citrate group; the DM + I/R and DM + I/R + Curcumin groups showed an increase in corporal weight during the 2nd week, 3rd week, 4th week when compared to the Citrate group, as shown in Table 2. The DM + I/R group presented a slight variation in this parameter in the first and fourth weeks when compared to the Citrate and the DM group.

An increase in the kidney weight in the DM and DM + I/R groups was observed when compared to the Citrate group, which did not occur in the DM + I/R + Curcumin group.

Additionally, the kidney weight and animal weight ratio of the untreated diabetic groups were higher than in the Citrate group. The DM + I/R + Curcumin group had significantly altered parameters when compared to the DM group, as shown in Table 3.

Table 4 shows that the DM group presented a higher food and water intake only when compared to the Citrate group. An increase in the pattern of food and water intake was noticed also in the DM + IR + Curcumin group when compared to the DM and Citrate groups. The DM + I/R group showed an increase in food intake when compared to the Citrate and DM groups, and an increase in water intake when contrasted with the DM group. Curcumin administration did not interfere with these parameters significantly.

### 3.2. Renal Function

Renal function was evaluated according to the urinary flow parameters, serum creatinine, inulin clearance and NGAL measurement. Citrate group values were considered as a normal reference.

The DM and DM + I/R + Curcumin groups presented an increase in urinary flow compared to Citrate, but the DM and DM + IR groups showed a reduction in this parameter. DM + I/R + Curcumin presented an increase in urinary flow close to normal levels.

As for the measurement of serum creatinine, an increase was observed in the diabetic groups compared to the Citrate group. The DM + I/R group showed an additional increase in this parameter when compared to the DM and Citrate group. Curcumin treatment reduced this variable compared to the untreated DM + I/R group.

Additionally, a reduction in inulin clearance in the diabetic groups was observed when compared to Citrate. This reduction was more accentuated in the DM + I/R group compared to the other DM groups. The DM + I/R + Curcumin group showed an improvement in this variable compared to the DM + I/R group.

Finally, an increase in the measured values of NGAL was detected in the animals of the diabetic groups in relation to the healthy control group and was worse in the DM + I/R group. The improvement of this biomarker was seen after treatment with curcumin in the DM + IR + Curcumin group as shown in Table 5.

### 3.3. Global and Renal Hemodynamics

The assessment of renal hemodynamics was performed by checking the RBF and RVR considering the MAP and HR. The Citrate group was considered a normal control. HR and MAP showed a slight variability between groups, but with no statistical significance.

There was a reduction in RBF in DM, DM + I/R and DM + I/R + Curcumin animals compared to the Citrate group, while the DM + I/R group presented the most compromised RBF values in comparison with the DM. Additionally, curcumin administration promoted an increase in RBF when the DM + I/R + Curcumin group was compared to DM + I/R

The DM, DM + I/R and DM + I/R + Curcumin animals presented an increased RVR when compared to the Citrate group. The I/R insult worsened the RVR values compared to the DM group. The animals in the DM + I/R + Curcumin group showed a decrease in RVR values compared to the DM + I/R group, as shown in Table 6.

### 3.4. Oxidative Profile

The Citrate group presented results considered as reference for the oxidative parameters analyzed below.

The diabetic groups showed an increase in the excretion of urinary peroxides when compared to the Citrate group. The induction of I/R caused an additional increase in this parameter in relation to the DM group. Curcumin administration reduced peroxide excretion when DM + I/R + Curcumin was compared to DM + I/R and DM + I/R.

The DM, DM + I/R and DM + I/R + Curcumin animals showed an increase in TBARS compared to the Citrate group. The DM + I/R group showed a significant increase in this variable when compared to the DM group. This change was significantly lower in the DM + I/R + Curcumin group.

The diabetic groups showed a noticeable increase in NO, demonstrated by the increase in urinary nitrate when compared to the Citrate group. The DM + I/R group showed an increase in this parameter when compared to the DM group. Curcumin administration reduced NO excretion when the DM + I/R + Curcumin group was compared to the DM + I/R group.

The consumption of thiols increased in the diabetic groups when compared to the Citrate animals. The DM + I/R group demonstrates that the I/R insult caused greater consumption of the antioxidant reserve than in the DM group. The DM + I/R + Curcumin group showed an increase in thiols compared to the DM + I/R group as shown in Table 7.

## 4. Discussion

Complications in renal function, renal hemodynamics and oxidative profile induced by DM were reproduced in the experimental model of STZ-induced diabetes. Additionally, the successful performance of the acute insult by I/R in diabetic animals and the renoprotective role of curcumin in this model were confirmed by the results.

The presence of DM is configured as a risk factor for the occurrence of acute I/R insult [1]. The I/R causes a reduction in renal function with volume impairment, causing tubular and vascular lesions [25,26]. As shown in the present study, diabetic animals undergoing I/R had additional impairment of renal function, demonstrated by the reduction of inulin clearance, a marker of renal function considered the gold standard in basic research. Furthermore, NGAL values, an early biomarker of tubular and structural nephron damage, were shown to be sensitive to AKI, presenting altered values, especially after the I/R process [27,28].

Additionally, serum creatinine, a clinical marker of renal dysfunction, showed an increase, corroborating the findings in other experimental studies with the same acute injury model [13,26]. Animals submitted to I/R had an increase in oxidative and nitrosative metabolites (peroxides, TBARS, urinary nitrate) and presented impairment of renal hemodynamics (reduced renal blood flow and increased renal vascular resistance). The hemodynamic changes, which include the decrease in RBF and the increase in RVR, observed in the diabetic animals in the present study confirm the dysfunctions resulting from DM. When these are added to the I/R injury, their deleterious effects are accentuated with higher RVR and lower RBF [13].

Early treatment with curcumin, a phytomedicine with antioxidant pharmacological properties, interfered in the renal deleterious effects present in DM, which is known to compromise renal hemodynamics by decreasing RBF and increasing RVR, as described above. Furthermore, treatment with curcumin induced beneficial results by decreasing the elimination of urinary peroxides, TBARS and preserving thiol reserves in the renal tissue [28].

Current evidence has revealed that curcumin has pharmacological effects in the prevention and mitigation of various diseases, including kidney disease [24]. A study with STZ diabetic rats showed a reduction in glycemic levels, serum creatinine, urea nitrogen and albuminuria [29]. Fatih et al. demonstrated, in a preclinical I/R study, that the use of curcumin reduced inflammatory process, oxidative stress and apoptosis by the semaphorin–plexin [30]. In a translational view, a meta-analysis of randomized, double-blind, placebo-controlled clinical trials about curcumin supplementation in patients with diabetic kidney disease (DRD) showed a significant improvement in serum creatinine corroborating the findings of this investigation [31,32].

It is also well known that the oxidative stress induced by STZ administration is followed by an inflammatory reaction, and the first cell type that contributes to cellular response and infiltrates the islet cells are the macrophages [33]. The production of cytokine is related to the development of diabetes and, due to its cytotoxic properties for beta islet cells, STZ can induce DM without involving the autoimmune mechanism [34]. Thus, the cytotoxicity of STZ is presumed to be mediated by reactive oxygen species (ROS), reactive nitric oxide species, and induction of inflammatory responses [35]. Furthermore, hyperglycemia leads to more elevated ROS, which has an essential effect on some metabolic pathways and promotes diabetic vascular disease [36]. Considering that the early pathophysiological mechanisms of STZ diabetes are related to oxidant injuries, this study could confirm the antioxidant properties of curcumin in preventing the kidney disturbances in DM rats submitted to I/R additional damage.

This study reinforces the importance of prevention as the most effective method in health, and as already foreseen in the National Policy on Integrative and Complementary Practices, the use of phytomedicines is presented as an aid to the prevention or treatment of diseases such as DM.

Therefore, this study proved the effectiveness of curcumin as a renoprotective agent in DM with I/R, becoming a promising agent for the suppression of the morbidity of DM. In addition, the study suggests that, after other preclinical studies with different formulations and administration routes designed to obtain products with higher bioavailability, with a larger sample and other methodologies for evaluating renal function and also clinical studies, curcumin can be considered a therapeutic possibility to be incorporated into the clinical treatment of diabetic patients at risk for complications of renal function.

## 5. Conclusions

Early treatment with curcumin improved renal function in diabetic rats submitted to I/R with beneficial repercussions on renal hemodynamics and renal oxidative profile.

## Figures and Tables

**Table 1 nutrients-14-02798-t001:** Monitoring of blood glucose.

Groups	*n*	Initial—48 h	1st Week	2nd Week	3rd Week	4th Week
Citrate	5	86.7± 3.7	89.7 ± 8.3	97.3 ± 10.7	86.2 ± 11.6	95.7 ± 7.7
DM	5	374.6 ± 51.3 ^a^	389.3 ± 76.2 ^a^	389.7 ± 91.2 ^a^	380.7 ± 84.0 ^a^	390.0 ± 58.0 ^a^
DM + I/R	5	342.6 ± 57.4 ^a^	342.0 ± 67.9 ^a^	334.7 ± 37.5 ^a^	374.6 ± 78.1 ^a^	345.9 ± 57.2 ^a^
DM + I/R + Curcumin	5	330.7 ± 63.3 ^a^	349.4 ± 43.3 ^a^	346.4 ± 55.6 ^a^	335.1 ± 66.6 ^a^	311.3 ± 66.5 ^a^

^a^
*p* < 0.05 versus Citrate; ^b^ *p* < 0.05 versus DM; ^c^ *p* < 0.05 versus DM + I/R.

**Table 2 nutrients-14-02798-t002:** Body weight.

Groups	*n*	1st Week	2nd Week	3rd Week	4th Week
Citrate	5	278 ± 57	306 ± 60	343 ± 49	366 ± 54
DM	5	281 ± 9	274 ± 10	272 ± 13 ^a^	247 ± 16 ^a^
DM + I/R	5	216 ± 22 ^a,b^	233 ± 35 ^a^	243 ± 42 ^a^	281 ± 17 ^a,b^
DM + I/R + Curcumin	5	258 ± 14	284 ± 23	279 ± 18 ^a^	309 ± 14 ^a^

^a^
*p* < 0.05 versus Citrate; ^b^ *p* < 0.05 versus DM; ^c^ *p* < 0.05 versus DM + I/R.

**Table 3 nutrients-14-02798-t003:** Kidney weight and body weight/animal weight ratio.

Groups	*n*	Kidney Weight (g)	Kidney Weight/Animal Weight
Citrate	5	1.3 ± 0.1	0.3 ± 0.1
DM	5	1.9 ± 0.2 ^a^	0.6 ± 0.04 ^a^
DM + I/R	5	1.8 ± 0.4 ^a^	0.5 ± 0.1 ^a^
DM + I/R + Curcumin	5	1.4 ± 0.2	0.5 ± 0.1 ^a,b^

^a^
*p* < 0.05 versus Citrate; ^b^ *p* < 0.05 versus DM; ^c^ *p* < 0.05 versus DM + I/R.

**Table 4 nutrients-14-02798-t004:** Food and water intake.

Groups	*n*	Food (g)	Water (mL)
Citrate	5	23.0 ± 1.7	22.5 ± 2.9
DM	5	35.4 ± 0.9 ^a^	88.0 ± 13.0 ^a^
DM + I/R	5	19.8 ± 9.4 ^a,b^	35.0 ± 20.0 ^b^
DM + I/R + Curcumin	5	20.7 ± 8.3 ^a,b^	59.2 ± 15.7 ^a,b^

^a^
*p* < 0.05 versus Citrate; ^b^ *p* < 0.05 versus DM; ^c^ *p* < 0.05 versus DM + I/R.

**Table 5 nutrients-14-02798-t005:** Renal function.

Groups	*n*	24 h Urinary Flow (mL/min)	Serum Creatinine (mg/dL)	Inulin Clearance (mL/min)	Urinary NGAL (ng/mL)
Citrate	5	0.011 ± 0.003	0.28 ± 0.05	0.91 ± 0.26	41.41
DM	5	0.056 ± 0.010 ^a^	1.08 ± 0.14 ^a^	0.58 ± 0.04 ^a^	57.25
DM + I/R	5	0.011 ± 0.005 ^b^	2.76 ± 0.67 ^a,b^	0.15 ± 0.06 ^a,b^	142.42 ^a,b^
DM + I/R + Curcumina	5	0.025 ± 0.005 ^a,b,c^	0.93 ± 0.12 ^a,c^	0.44 ± 0.11 ^a^	81.35 ^c^

^a^
*p* < 0.05 versus Citrate; ^b^ *p* < 0.05 versus DM; ^c^ *p* < 0.05 versus DM + I/R.

**Table 6 nutrients-14-02798-t006:** Renal hemodynamics.

Groups	*n*	Heart Rate (Beats per min)	Mean Arterial Pressure (mmHg)	Renal Blood Flow (mL/min)	Renal Vascular Resistance (mmHg/mL/min)
Citrate	5	460 ± 56	97 ± 9	8.1 ± 1.3	11.3 ± 1.9
DM	5	471 ± 23	118 ± 26	4.8 ± 0.4 ^a^	26.7 ± 6.8 ^a^
DM + I/R	5	500 ± 41	95 ± 5	2.2 ± 0.3 ^a,b^	43.5 ± 5.7 ^a,b^
DM + I/R + Curcumin	5	522 ± 29	99 ± 12	5.1 ± 1.1 ^a,c^	20.9 ± 2.0 ^a,c^

^a^
*p* < 0.05 versus Citrate; ^b^ *p* < 0.05 versus DM; ^c^ *p* < 0.05 versus DM + I/R.

**Table 7 nutrients-14-02798-t007:** Oxidative profile.

Groups	*n*	Urinary Peroxides (nmol/g of Urinary Creatinine)	Lipid Peroxidation (nmol/g of Urinary Creatinine)	Urinary Nitrate (µM/g Urinary Creatinine)	Renal Tissue Thiols (nmol/mg of Total Protein)
Citrate	5	0.90 ± 0.12	0.29 ± 0.05	22.01 ± 5.84	25.98 ± 2.87
DM	5	3.59 ± 0.38 ^a^	10.45 ± 0.46 ^a^	51.70 ± 10.45 ^a^	15.52 ± 2.45 ^a^
DM + I/R	5	5.85 ± 0.25 ^a,b^	23.29 ± 2.30 ^a,b^	165.83 ±13.56 ^a,b^	11.01 ± 2.16 ^a,b^
DM + I/R + Curcumin	5	1.80 ± 0.48 ^a,b,c^	9.13 ± 2.40 ^a,c^	60.51 ± 10.87 ^a,c^	19.86 ± 2.77 ^a,c^

^a^
*p* < 0.05 versus Citrate; ^b^ *p* < 0.05 versus DM; ^c^
*p* < 0.05 versus DM + I/R.

## Data Availability

The data presented in this study are available on request from the corresponding author.

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
