# Peer review of "The Effect of Curcumin on Renal Ischemia/Reperfusion Injury in Diabetic Rats"

_nutrients, 2022, doi:10.3390/nu14142798_

Round 1
Reviewer 1 Report
Diabetes is an important health problem that affects millions of people each year around the world. People who have diabetes often have kidney problems as the workload on the kidney increases during diabetes. Machado Di et al have shown that curcumin can be used to alleviate the kidney functions in diabetic rats. Authors have a good model to study diabetes and kidney injury during diabetes. But unfortunately the. Study is not very well executed and also similar ideas were tried with curcumin already (PMID: 30741618).
Comments:
1. There are too many typos in the documents.
2. How curcumin acts as a renoprotective agent? Is it through antioxidative or anti-inflammatory effects? It is not discussed well in the manuscript.
3. Curcumin is a well known anti-inflammatory compound, but this has not been tested in this study context.
Author Response
Thank you for you considerations. Ajustments includede in lina 64 to 71 and 323 to 329 as following
Curcumin has shown to be effective in the regulation of glycaemia and lipidemia [9] with an action mechanism similar to antidiabetic medications [10].
In addition to that, recent studies have demonstrated promising effects of curcumin for the treatment of autoimmune diseases, such as type 1 diabetes [8]. It has been also demonstrated that it could have a potential role in preventing and treating several diseases due to its anti-bacterial, anti-viral, anti-ischemic, hepato-protective, nephron-protective, antirheumatic, and anticancer activities.
Early treatment with Curcumin, a phytomedicine with antioxidant pharmacological properties, interfered in the renal deleterious effects present in DM, which is known to compromise renal hemodynamics by increasing RBF and decreasing RVR, as described above. Furthermore, treatment with Curcumin induced beneficial results by decreasing the elimination of urinary peroxides, TBARS and preserving thiol reserves in the renal tissue
Dear reviewer, in response to your comments to enrich our manuscript, review and corrections were carried out by the authors where it was improved:
The introduction with relevant information including its references;
We improve the references bringing references that support our findings;
As well as we improved the way to show our results by tables and not graphics and spelling errors were improved. In addition, we were able to demonstrate more clearly the antioxidant effects of curcumin in our research model, which was the causative agent of renoprotection.

Reviewer 2 Report
In this study authors evaluated the effect of Curcumin in diabetic rats with AKI and found that IR worsen renal function and oxidative stress in DM rats, but pretreatment with Curcumin improved inulin clearance, decreased serum creatinine and NGAL. Moreover, renal hemodynamics of this group was improved and, oxidative and nitrosative metabolites were reduced.
This manuscript is very intereting but some points must be improved. In particular, manuscipt presents to many heading and subheading that makes reading very difficult. Moreover:
- Lines 53-65: Although the authors highlighted anti-inflammatory and antioxidants effects of curcumin, It deserves to be pointed out that curcumin plays an important role in ameliorating/preventing Diabetes Mellitus and Gestational Diabetes Mellitus (PMID: 33477354, 31906278). This point is very important since the aim ot this study regards the role of curcumin in this pathology.
- 2. Objective: I would suggest to remove this paragraph to avoid excessive subheading in the manuscript. This is just the aim of the study and can be insert in the introduction.
- 3. Materials and Methods: Avoid so many subheading because it makes reading difficult
- 3.5.3. Urinary NGAL: Please insert the product code
- 5. Results: Graphs are very difficult to read, please improve image quality. Moreover, these are not tables but figures, please replace.
- Authors must follow journal style
Author Response
Obrigado por suas considerações. Ajustes incluídos na linha 64 a 71 e 323 a 329 conforme segue:
A curcumina tem se mostrado eficaz na regulação da glicemia e lipidemia [9] com mecanismo de ação semelhante aos medicamentos antidiabéticos [10] .
Além disso, estudos recentes demonstraram efeitos promissores da curcumina para o tratamento de doenças autoimunes, como diabetes tipo 1 [8] . Também foi demonstrado que pode ter um papel potencial na prevenção e tratamento de várias doenças devido às suas atividades antibacteriana, antiviral, anti-isquêmica, hepatoprotetora, protetora de néfrons, antirreumática e anticancerígena.
O tratamento precoce com Curcumina, um fitoterápico com propriedades farmacológicas antioxidantes, interferiu nos efeitos deletérios renais presentes no DM, que sabidamente compromete a hemodinâmica renal por aumentar o FSR e diminuir o RVR, conforme descrito acima. Além disso, o tratamento com Curcumina induziu resultados benéficos ao diminuir a eliminação de peróxidos urinários, TBARS e preservar as reservas de tiol no tecido renal
Prezado revisor, em resposta aos seus comentários para enriquecer nosso manuscrito, a revisão e correções foram realizadas pelos autores onde foi aprimorado:
A introdução com informações relevantes incluindo suas referências e objetivo na linha 76 a 77. Materiais e métodos foi retirado os subtitulos, além de ser inserido o cód do kit de NGAL utilizado nessa pesquisa e os resultados foram substituidos por tabelas respeitando o modelo da revista como sugerido.

Round 2
Reviewer 1 Report
Dear Authors,
1. I do still see a lot of typos in the manuscript. I appreciate if you take care of it.
2. I would have liked to see the manuscript with bar diagrams with P values, showing significance among the groups rather than tables.
I think manuscript can be accepted after these minor changes
Reviewer 2 Report
manuscript has been significantly improved and can be accepted in the present form